# Infection and Prevention of Rabies Viruses

**DOI:** 10.3390/microorganisms13020380

**Published:** 2025-02-09

**Authors:** Shiu-Jau Chen, Chung-I Rai, Shao-Cheng Wang, Yuan-Chuan Chen

**Affiliations:** 1Department of Neurosurgery, Mackay Memorial Hospital, Taipei 10449, Taiwan; chenshiujau@gmail.com; 2Department of Medicine, Mackay Medical College, New Taipei City 25245, Taiwan; 3Department of Cosmetic Science, Vanung University, 1 Van Nung Road, Chung-Li City, Taoyuan 320676, Taiwan; barry.rai@gmail.com; 4Department of Psychiatric, Taoyuan General Hospital, Ministry of Health and Welfare, Taoyuan 33004, Taiwan; 5Department of Psychiatry, Kaohsiung Chang Gung Memorial Hospital and Chang Gung University College of Medicine, Kaohsiung 833, Taiwan; 6Department of Psychiatry, Kaohsiung Municipal Ta-Tung Hospital, Kaohsiung 801, Taiwan; 7Department of Nursing, Jenteh Junior College of Medicine, Nursing and Management, Miaoli County 356006, Taiwan; 8Department of Medical Technology, Jenteh Junior College of Medicine, Nursing and Management, Miaoli County 356006, Taiwan

**Keywords:** rabies, pre-exposure, post-exposure, pre-exposure and post exposure vaccination, purified Vero cell-cultured freeze-dried rabies vaccine, mRNA-based vaccine

## Abstract

Rabies is a fatal zoonotic disease and causes about 59,000 human deaths globally every year. Especially, its mortality is almost 100% in cases where the rabies virus has transmitted to the central nervous system. The special virus life cycle and pathogenic mechanism make it difficult for the host immune system to combat rabies viruses. Vaccination including pre-exposure and post-exposure prophylaxis is an effective strategy for rabies prevention. The pre-exposure vaccination is mainly applied for animals and the post-exposure vaccination is the most application for humans. Although rabies vaccines are widely used and seem to be safe and effective, there are some disadvantages, limitations, or challenges affecting vaccine promotion and distribution. Therefore, more effective, convenient, safer, and cheaper rabies vaccines have been developed or are being developed. The development of novel human rabies vaccine is mainly focusing on vaccines based on a purified Vero cell-cultured freeze-dried rabies vaccine (PVRV). PVRV has been demonstrated to be promising to make the rabies vaccine more effective and secure in animal studies or clinical trials. Moreover, mRNA-based vaccines have been shown to have the potential to enhance the safety and efficacy of rabies vaccines for both animal and human uses.

## 1. Infection of Rabies Virus

The rabies virus, a member of the *Lyssavirus* genus of the *Rhabdoviridae* family, is a neurotropic virus that causes a serious zoonotic disease called rabies [1]. Different from most viruses, it spreads through neurons, not blood or lymph in humans and animals. Rabies is a fatal viral infection that is mainly transmitted through bites, scratches or licks of broken skin or mucous membranes from infected animals. The mortality of rabies is around 100% once rabies viruses have invaded the central nervous system (brain and spinal cord, CNS) [1].

### 1.1. Molecular Biology

Rabies viruses have helical symmetry and their infectious particles are cylindrical shape like a bullet or a stick. This virus has an envelope and its genome consists of a single-stranded linear RNA (ssRNA) with negative-sense, containing 11,615~11,966 nucleotides [1,2,3,4,5]. The genetic information is packaged as a ribonucleoprotein complex in which ssRNA is bound with the nucleoprotein (N). The viral RNA genome includes five highly conserved genes which encode nucleoprotein (N), phosphoprotein (P), matrix protein (M), glycoprotein (G) and large structure protein (RNA replicase) (L) [6]. The L protein interacts with the P protein to generate an RNA-dependent RNA polymerase complex. The replication and transcription events of rabies viruses take place inside a specialized compartment (factory) termed Negri body in the cytoplasm [7,8,9]. The Negri body, an eosinophilic inclusion body formed by the accumulation of viral capsid proteins in cells, is widely distributed in the CNS cells of patients and causes main damage in host cells [7,8,9]. Negri bodies are typical for the infection of rabies viruses and used as an indicator for rabies diagnosis [4] (Figure 1).

### 1.2. Pathogenic Mechanism

Following the G protein binds with the nicotinic acetylcholine receptor (nAChR) on host skeletal neuromuscular junction, rabies viruses enter host cells by the endosomal transport pathway. To date, other receptors including neural cell adhesion molecules (NCAM), p75 neurotrophin receptor (p75NTR), metabotropic glutamate receptor 2 (mGluR2), carbohydrates, and gangliosides have been identified [10]. The G protein fuses with the host cell membrane to form endophagic vesicles and enter the cytoplasm, and then undergo uncoating, transcription, replication, assembly, and finally release from host cells through budding. Inside the endosome, the membrane fusion is triggered by the low pH value, thus promoting the viral genome to reach the cytosol. Both receptor binding and membrane fusion are catalyzed by the G protein, which plays a key role for pathogenesis [11,12,13]. Initially, they only replicate in skeletal muscle or skin cells, but gradually release and enter the axon through the motor end plates at the neuromuscular junction. Later, they rapidly ascend through retrograde axonal transport to infect neurons through nerve endings at a speed of 5–100 mm per day [14]. It takes variable time (usually several days, one month or longer period) to spread to the CNS, depending on the distance from the bite lesion sites to the CNS. P protein can acts as an interferon antagonist throughout a replication period of rabies virus in infected cells to reduce the host innate immunity to against invading viruses [15,16]. Finally, rabies viral particles invade to the CNS through the retrograde axonal transport and spread more rapidly, accompanied by severe physiological changes and pathological phenomena, causing the necrosis of central neurons and lesions of glial cells, which are the major pathogenesis steps after infection [17]. However, the definite mechanism of the retrograde axonal transport is still poorly understood. Furthermore, the rabies virus moves to other organs especially the salivary glands from the CNS, thereby making viruses to be transmitted from infected animals to humans through salivation [18,19].

Some studies have shown that the expression of interferons, cytokines, and chemokines was significantly stimulated only when the rabies virus invade into the CNS, and thereby the transcription levels induced by interferons and cytokines were increased [20,21,22]. The increased expression of chemokines in nerve cells promotes immune responses, especially T cells enter the CNS. It is not until the virus enters the CNS that the immune system detects viral invasion and begins to respond. However, it is not clear how the rabies virus escapes immune regulation in surrounding tissues [23,24,25,26]. Maybe, the level of viral initial replication is so low that the immune system cannot detect the presence of the rabies virus in the early stages of infection [23,24,25,26]. The other possible mechanism is that the surrounding immunity is suppressed by rabies virus and failed to initiate immune responses in time to fight the invading rabies viruses [23,24,25,26]. The recent study also suggested that host *gasdermin D* (*Gsdmd*) genes related to the pyroptosis (a highly inflammatory form of lytic programmed cell death that occurs often upon an intracellular rabies virus infection) pathway were significantly upregulated to cause the neuron damage [27].

### 1.3. Signs and Symptoms

The rabies virus attacks the CNS and symptoms gradually worsen as the disease progresses. The CNS damage caused by rabies viruses will cause paralysis and patients will experience depression, coma, abnormal breathing, and finally death when it invades the spinal cord and brainstem. The symptoms of rabies can mainly be divided into three stages in humans or animals as follows [10,28,29,30,31]: (1) No symptom (incubation period): This period is usually 1 to 3 months, but it can be as short as a few days or as long as years. The length of the incubation period is variable, depending on the site of the bite (the closer to the brain, the shorter the incubation period), patient immune responses, and the amount of virus. (2) Prodromal symptoms (initial symptoms): Symptoms start to appear after incubation period, including fever, tiredness, headache, tingling, numbness, burning at the injured site, malaise, loss of appetite, and sometimes flu-like symptoms. The duration of this period is 2–10 days. (3) Nervous system symptoms (attack period): Rabies patients will develop neurological symptoms after entering the attack phase, which are usually divided into the furious and the paralytic form. The more one is furious (excited) rabies (about 80% of cases). Its symptoms include extreme agitation, hydrophobia (seeing or hearing water running can trigger convulsions), aerophobia (hypersensitivity to wind or air currents, even air movement can cause pain), inability to swallow, excessive drooling, hallucinations, excessive excitement, aggression, and abnormal behavior. This phase may last for hours or days, after which patients begin to have an excited episode rather than constant agitation [32,33]. The less one is paralytic (quiet) rabies (about 20% of cases). Its symptoms include muscle weakness, gradual paralysis (usually starts at the site of the bite), a gradual slip into coma and eventually death. Patients with paralytic rabies are relatively quiet and have a slow progression of symptoms. The course of the paralytic form is a little more prolonged, beginning with tingling or paralysis of the bitten limb nearest the wound and extending upwards to cause quadriplegia [32,33].

### 1.4. Host, Reservoir, and Vector

The hosts of rabies viruses include any mammals, as the virus can infect and cause disease in all mammalian species. Common hosts include domestic animals like dogs, cats, and livestock (e.g., cattle, horses, sheep, etc.). Wild animal hosts include bats, raccoons, foxes, skunks, and mongooses [34,35,36,37].

Rabies reservoirs are species that maintain and transmit the virus in a given region without succumbing to viral infection at high rates. These animals are critical for the persistence of the rabies virus in nature [34,35,36,37]. Bats are globally important reservoirs which carry diverse rabies viruses everywhere. Dogs are primary reservoirs and responsible for most human cases through bites in Asia and Africa. Wild animals including raccoons, skunks, foxes, and bats are primary reservoirs in North America, Europe, and Australia. Bats (mainly *Desmodus rotundus*, the common vampire bat) and sometimes wild carnivores like foxes are primary reservoirs in Latin America [34,35,36,37].

The transmission of rabies viruses in domestic dogs has been known for centuries, but bats and wild carnivores began to be recognized as reservoirs in early 20th century. Bats are ancestral reservoirs of rabies viruses, and their diversified species perpetuate distinct viral lineages [38]. The circulation of rabies viruses in domestic dogs maybe a historic shift from bat reservoirs. Rabies viruses had repeated host shifts to wild carnivores around the world since they were established in the domestic dog population [38]. While reservoirs are often vectors, vectors can include any mammals infected with rabies viruses. The major vectors are illustrated as follows [39,40]: (1) Dogs: the primary transmitters of rabies to humans, accounting for over 99% of human rabies deaths globally. (2) Domestic animals: livestock or cats, in rare cases. Although cats are important vectors of lyssaviruses, they are only incidental hosts and not viral reservoirs [41]. (3) Wildlife: bats, foxes, raccoons, and skunks.

### 1.5. Epidemiology and Control

Rabies is mainly transmitted from rabid animals to humans through bites or licks. Rabies is extensively spread in developing countries in Asia, Africa, and Latin America, especially in areas where rabies control among dog populations is poor. According to data from the World Health Organization (WHO), Asia and Africa account for more than 95% of the global annual rabies deaths [33]. Rabies is estimated to cause approximately 59,000 deaths in the world each year, with more than 90% of cases resulting from dog bites. Most deaths have occurred in areas with limited resources and vaccine shortages.

Currently, the species in the Lyssavirus genus are distinguished in two phylogroups: Phylogroup I includes the rabies virus, Australian bat lyssavirus (species *Lyssavirus australis*), Duvenhage virus (*Lyssavirus duvenhage*), European bat lyssavirus 1 (*Lyssavirus hamburg*), European bat lyssavirus 2 (*Lyssavirus helsinki*), Aravan virus (*Lyssavirus aravan*), Khujand virus (*Lyssavirus khujand*), Kotalathi bat lyssavirus (*Lyssavirus kotalahti*), Bokeloh bat lyssavirus (*Lyssavirus bokeloh*), Irkut virus (*Lyssavirus irkut*), Taiwan bat lyssavirus (*Lyssavirus formosa*), and Gannoruwa bat lyssavirus (*Lyssavirus gannoruwa*); phylogroup II includes Mokola virus (*Lyssavirus mokola*), Lagos bat virus (*Lyssavirus lagos*), and Shimoni bat virus (*Lyssavirus shimoni*). However, West Caucasian bat virus (*Lyssavirus caucasicus*), Ikoma lyssavirus (*Lyssavirus ikoma*), and Lleida bat lyssavirus (*Lyssavirus lleida*) are not members of either of these two phylogroups [42]. These species are different in terms of neurotropism, pathogenesis, the induction of apoptosis, immunogenicity, and on a molecular basis [42,43]. It is known that vaccination is one of the effective strategies to control rabies for both animals and humans. Since human rabies is mainly resulted from dog rabies, it is possible to decrease human rabies through the vaccination and population control of dogs. However, it represents a complex challenge to control lyssaviruses from bats today. Therefore, the bat rabies is a growing threat for both human and animal health, though the Lleida bat lyssavirus may not be as lethal as other viruses in some studies [44]. Because most bats can transmit various lyssavirus species, which the available vaccines cannot prevent in animals and humans, there is a need to broaden the protection spectrum of rabies vaccines to lyssavirus vaccines. The most abundant ribonucleoproteins in infected cells can cross-react among all members of the Lyssavirus genus to promote the use of standardized detection of all lyssaviruses. In contrast, G proteins are relatively conserved within phylogroups (ectodomain conservation > 75%) but not between phylogroups (ectodomain conservation < 65%) [42]. Consequently, current rabies vaccines and immunoglobulins that induce or provide neutralizing antibodies against G proteins only protect against the infection of phylogroup I but not other lyssaviruses [42]. The chimeric G proteins which fusion of two halves from different species are effective to elicit complete immune responses and extend the vaccine spectrum from against rabies virus to lyssavirus [43]. Moreover, the lyssavirus G protein can act as epitopes or antigens for the development multivalent vaccines against various zoonoses [43]. The pre-fusion trimeric conformation of G protein demonstrates epitopes bound by protective neutralizing antibodies that induced by vaccination or passive administration for post-exposure prophylaxis [45]. Additionally, mRNA vaccine encoding G proteins can improve cross-neutralization of correlated lyssavirus strains [46], showing this platform has potential to develop a extensively protective vaccine against these lyssaviruses including phylogroup I, II, and members of other phylogroups than these two.

## 2. Vaccines for the Prevention of Animal Rabies

Rabies is still prevalent and underestimated particularly in undeveloped or developing countries, because human healthcare is not sufficient and domestic dogs have not been widely vaccinated [47]. Though rabies is also transmitted by wild animals (e.g., bats, raccoons, skunks, and foxes), and larger carnivores are considered rabid if they unforeseeably attack a person, pets and dogs are the main vectors to transmit rabies to humans if they are not vaccinated [47]. Since someone cannot avoid contact with pets, dogs, and other domestic animals, and it is also unrealistic to exclude the possibility that these animals may contact with rabid vectors, the vaccination of these animals for pre-exposure prophylaxis is the best strategy to prevent animal rabies. The transmission of rabies can be effectively reduced and humans can be protected from the threat of the disease through the extensive vaccination of pets, dogs, and domestic animals, especially dogs. The public health organizations usually stipulate laws or standards for animal vaccination against rabies to ensure the health and safety of humans and animals.

### 2.1. Traditional Vaccine

The first rabies vaccine, which was a dried homogenate of rabbit spinal cord and an inactivated vaccine, was developed by Louis Pasteur in 1885. Dr. Pasteur also devolved a live attenuated vaccine, which was a homogenate of one-day-dried spinal cord likely including live rabies viruses. In the following decades, the scientists gradually improved the preparative methods and technologies of rabies vaccines. In the 1950s, inactivated rabies vaccines based on chicken embryo cells began to be widely applied. The rabies vaccines in this period were also used in humans, especially for emergency prevention or treatment after rabies exposure. The commonly used animal rabies vaccines are mainly divided into three categories: one is an inactivated vaccine, another is a live attenuated vaccine, and the other is a recombinant vaccine (Table 1). Historically, inactivated vaccines have been widely used for domestic and companion animals, and live attenuated vaccines are usually used for controlling rabies in wild animals.

### 2.2. Traditional Vaccine Challenges

The vaccination program for animal rabies is usually formulated according to the species, age, health status, and exposure risk. For common pets or domestic dogs, the recommended age for initial rabies vaccination is at 6–8 weeks of age, then every 2–4 weeks until 16 weeks of age or older, and booster vaccination is given either at 12 months of age or 12 months after the last of the primary series of vaccines [58]. In some areas with high incidence of rabies, more frequent vaccinations may be required. The vaccination strategies for animal rabies are as follows: (1) Pre-exposure immunization: The pets, dogs or domestic animals are regularly vaccinated to prevent the rabies and ensure the immunization coverage. (2) Post-exposure immunization: After the animal is exposed to a suspected rabid animal, it is required to have timely vaccination for prevention. (3) Oral vaccination: Oral vaccination of wild animals (e.g., foxes, raccoons, etc.) to reduce the natural spread of rabies. Although the effectiveness of rabies vaccines has been well proven in animals, the rate of animal vaccination is still not enough worldwide, and its actual protection efficiency is difficult to evaluate due to some limitations (Table 2).

### 2.3. Development of Novel Animal Rabies Vaccines

The animal rabies vaccine plays an important role in controlling the spread of rabies virus, and thus become the core tool for the prevention of rabies. With the development of vaccine technology, current animal rabies vaccines have been significantly improved in terms of immune efficacy, safety, and production efficiency. However, vaccination coverage and global immunization strategies still need to be further promoted. The threat of rabies can finally be eliminated only through global cooperation, the increment of vaccination rates, and the execution of effective control strategies. The research and production methods of animal rabies vaccines have tremendous changes with the advancement of biotechnologies and immunology. Some novel vaccines have been developed or being developed recently.

#### 2.3.1. Inactivated Rabies Vaccine Using Adjuvants

In 2023, Yu et al. used PIKA which was a stabilized double-stranded RNA that interacts with toll-like receptor-3 (TLR-3) as an adjuvant to enhance rabies immunization [70]. They found that this PIKA rabies vaccine showed over 80% protective efficacy without immunoglobulin in mice infected with seven rabies strains prevalent in China [70]. The PIKA rabies vaccine elicited more neutralizing antibody just 5 days post-vaccination, compared to by licensed rabies vaccines [70]. The PIKA rabies vaccine also significantly resulted in more T cells that produce IFN-γ in response to the antigen. Additionally, the levels of IL-1β, IL-6, CCL-2, and TNF-α were increased at the injection site [70]. The levels of chemotactic proteins and pro-inflammatory molecules were elevated in the serum. They further confirmed the mechanism of PIKA by the tests in TLR3-knockout mice, suggesting its function is dependent on the TLR3 pathway [70]. The study shows that the PIKA rabies vaccine is potentially employed as a significantly effective rabies vaccine.

In 2024, Sokol et al. evaluated if the protective activity of the inactivated rabies vaccine would be enhanced by the addition of an adjuvant based on recombinant bacteria *Salmonella typhimurium* flagellin in mice [71]. They used a series of inactivated dry culture vaccines for dogs and cats “Rabikan” (strain Shchelkovo-51) by adding an adjuvant at various dilutions to undergo animal studies [71]. The similar series of inactivated dry culture vaccine without an adjuvant was used as the control. They evaluated the protective activity of the inactivated rabies vaccine preparations by the NIH potency test [71]. The results showed that the value of specific activity of the rabies vaccine co-administered with the adjuvant was significantly higher (48.69 IU/mL), compared to the vaccine without adding the adjuvant (3.75 IU/mL) [71]. This study suggests that the recombinant flagellin is potentially used as an effective adjuvant in the composition of rabies vaccines.

Also in 2024, Zhou et al. developed single injection vaccines to minimize the number of immunizations [72]. First of all, they designed the vaccine using the rabies virus G protein as an antigen. The dynamic layer-by-layer films as erodible coating was used to have multiply pulsatile releases of G protein as a time-controlled release system. They found that the single-injection vaccine induced more potent humoral and cellular immune responses, compared to the corresponding multi-dose ordinary vaccines [72]. Furthermore, they designed a second single injection vaccine using lentinan as an adjuvant. The results revealed that this single-injection vaccine also triggered more humoral and cellular immune responses, compared to the corresponding multi-dose ordinary vaccines [72]. The study suggests that the second single-injection vaccine elicits more immune responses and is more efficient on the inhibition of rabies viruses than the first one because lentinan is able to booster immune responses.

#### 2.3.2. Oral Attenuated Rabies Vaccine

In 2023, Megawati et al. evaluated the immunogenicity of local dogs in Bali, Indonesia after oral administration of the third-generation live attenuated rabies vaccine (viral strain SPBN GASGAS) [73]. The dogs received the oral rabies vaccine either directly or were provided an egg-flavored bait that contained a vaccine-loaded sachet by the researcher [73]. They compared the humoral immune responses with two further groups of dogs: One group received a parenteral inactivated rabies vaccine and the other group was an unvaccinated control. They bled these dogs, respectively, before vaccination and 27~32 days after vaccination, and then tested the blood samples for the presence of virus-binding antibodies [73]. They found that the seroconversion rate in the three groups of vaccinated dogs were similar: bait: 88.9%; direct-oral: 94.1%; parenteral: 90.9%; control: 0%. The level of antibodies between orally and parenterally vaccinated dogs were not significantly different [73]. The results showed that this third-generation live attenuated rabies vaccine could trigger enough immune responses, compared to a parenteral vaccine under field conditions [73]. This study demonstrates that the oral rabies vaccine has the potential to be an important tool to target hard-to-reach free-roaming dogs in the mass dog vaccination program. 

#### 2.3.3. mRNA-Based Rabies Vaccine

In 2022, Li et al. used an optimized mRNA vaccine construct (LVRNA001) expressing RABV-G and evaluated its immunogenicity and protective capacity in mice and dogs [74]. They found that LVRNA001 induced neutralizing antibody production and a strong Th1 cellular immune response in mice. LVRNA001 was also proved to provide protection against challenge with 50-fold lethal dose 50 (LD_50_) of rabies virus in both mice and dogs [74]. The protective efficiency assay showed that dosing interval (14 days) induced more antibody than 3- or 7-day intervals in mice [74]. Finally, they evaluated the survival rates of dogs receiving two 25 μg doses of LVRNA001 for post-exposure immunization against rabies virus, compared to five doses of inactivated vaccine over the course of three months [74]. The results revealed that the survival rate in the LVRNA001 group (100%) is significantly higher than the inactivated vaccine control group (33.33%) [74]. The study demonstrates that LVRNA001 induces strong protective immune responses in mice and dogs, suggesting mRNA vaccine can provide a new preventive strategy for rabies.

In 2023, Hellgren et al. used two doses of a lipid nanoparticle-formulated unmodified mRNA vaccine coding for the rabies virus glycoprotein G (RABV-G) to induce RABV-G specific plasmablasts and T cells in blood and plasma cells in the bone marrow in non-human primates [46]. Compared to two doses of the licensed rabies vaccine Rabipur^®^, the levels of RABV-G specific plasmablasts and T cells in animals were higher [46]. Additionally, the RABV-G binding and neutralizing antibody titers which the mRNA vaccine generates were both higher than Rabipur^®^, though the degree of somatic hypermutation and clonal diversity of the response are similar for the mRNA and Rabipur^®^ vaccines [46]. The cross-neutralization of related lyssavirus strains could be improved by higher overall antibody titers induced by the mRNA vaccine [46]. This study suggests that mRNA is promising to be a platform to develop a broadly protective vaccine against rabies viruses.

## 3. Vaccines for the Prevention of Human Rabies

### 3.1. Current Vaccine

Rabies is almost fatal once rabies viruses invade into CNS; therefore, the prophylaxis of viral infection is crucial. Previously, the most commonly used human rabies vaccine was an inactivated vaccine. It can prevent 99% of deaths if the infected person is administered by rabies vaccine promptly after exposure. There are two major vaccination strategies: pre-exposure prophylaxis and post-exposure prophylaxis. Pre-exposure vaccination can protect can protect domestic animals like dogs and other pets from rabies virus infection and thereby reduces the risk of infections for humans. Pre-exposure vaccination is also recommended for humans who come into regular contact with animals, such as veterinarians, animal rescuers, or field workers. Moreover, the people that live areas of difficult access and far from health centers in close contact with bat populations like countries in the Amazons have been implementing this pre-exposure vaccination to reduce human cases. However, the post-exposure vaccination is the focus strategy for common people and recommended for those who are being bitten or scratched by rabid animals. The wound should be cleaned immediately and followed by rabies vaccine administration. Currently, patients with high suspicion of rabies need prompt post-exposure immunoglobulin injections and vaccinations to prevent the virus from entering the CNS. Current rabies vaccines or immunoglobulins which are widely used to prevent rabies in humans and animals for post-exposure prophylaxis are mainly divided into three types (Table 3). Rabies cell culture vaccines (CCV) including human diploid cell culture rabies vaccine (HDCV), hamster kidney cell vaccines (HKCV), Vero cell vaccines and purified chick embryo cell culture rabies vaccines (PCECV) all confer protective immunity against infection with rabies viruses. The anti-rabies serum is a preparation containing the specific immunoglobulin. Rabies immunoglobulin is a medication made up of antibodies against the rabies virus and used to prevent rabies following exposure.

### 3.2. Current Vaccine Disadvantages

Vaccination is a good method for rabies control, but challenges are present, such as different safety profiles, immunogenicity among vaccine types, economic barriers to post-exposure prophylaxis, and limited education andcomprehensive awareness. Additionally, cold chain requirement may be an obstacle for the use of rabies vaccines. The rabies vaccine is widely used and effective for the prevention of rabies virus infection in humans, but there are some disadvantages affecting even hindering vaccine promotion and distribution including incomplete immune responses, side effects, multiple dose vaccination requirement, expense, insufficient vaccine supply, and cold chain requirements (Table 4).

### 3.3. Development of Novel Human Rabies Vaccines

Although human rabies vaccines have been used for years, the vaccination still fails to protect some infected patients in specific cases. In some resource-limited areas, it is still difficult to distribute and promote vaccination. Therefore, the development of new vaccines for the improvement of delayed antibody response and weak cellular immunity is needed to provide full protection [87]. Additionally, it is crucial to reduce the cost and enhance the convenience of vaccination. Recently, more effective, secure, thermostable, and cheaper human rabies vaccines have been developed or are being developed.

#### 3.3.1. Rabies Vaccines Based on Vero Cells

Vero cells (green monkey kidney cells) are a continuously aneuploidy cell line, indicating they have abnormal number of chromosomes. The continuous Vero cell line is known to have many division cycles without aging. Unlike other mammalian cells, Vero cells do not secrete interferon α/β when they are infected with viruses, because their interferon secretion genes are defective [90]. However, they still have interferon α/β receptors (IFNARs) and can respond to interferon as they are added to the culture medium. Vero cells have no animal or human components and low residual DNA contents. Therefore, they have been extensively used in research on the molecular mechanisms of viral infection and production of vaccines such as a purified Vero cell-cultured freeze-dried rabies vaccine (PVRV) [91]. Some successful clinical trials are illustrated as follows.

In 2022, Quiambao et al. developed a rabies vaccine candidate (PVRV-NG) in Vero cells to versus a licensed human diploid cell culture rabies vaccine (HDCV) in a pre-exposure regimen [92]. In a phase II randomized clinical study, healthy children aged 2–11 and adolescents aged 12–17 in the Philippines were randomized to receive three injections of either PVRV-NG or HDCV on day [D] 0, D7, and D28 by 2:1 allocation [92]. They measure the rabies virus-neutralizing antibodies at D0, D42, and 6 months (M6) after the first vaccination. They evaluated the safety during the vaccination period and up to 28 days posterior to the last injection. Additionally, they monitored serious adverse events until 6 months after last vaccination [92]. They found that all 342 participants including 171 children and 171 adolescents in both groups had a rabies virus-neutralizing antibody (RVNA) titer ≥0.5 IU/mL at D42, and more than 90% of participants had an RVNA titer ≥0.5 IU/mL at M6 [92]. This result revealed that PVRV-NG and HDCV are similar in their seroconversion rate. After each vaccination and up to 6 months following the last dose, the type and severity of adverse reactions were similar for both groups [92]. This study shows that the immune profile of PVRV-NG is almost the same as HDCV in a pre-exposure setting and well tolerated with no safety concerns, indicating PVRV-NG is promising to be a good human rabies vaccine.

In 2023, Pichon et al. developed a rabies vaccine candidate (PVRV-NG) in Vero cells, and this candidate was then reformulated to PVRV-NG2 [93]. They evaluated PVRV-NG in five multicenter, observer-blinded phase II trials and investigated the safety and immune response of three different doses (antigen content) of PVRV-NG2, compared to HDCV (Imovax Rabies^®^) [93]. In this trial, the healthy adults (*N* = 320) were randomized to receive PVRV-NG2 (low, medium, or high dose), PVRV-NG, or HDCV (2:2:2:1:1 ratio) in accordance with a five-dose post-exposure regimen on Day [D] 0, 3, 7, 14, and 28 by intramuscular (IM) injection; additionally, all participants received human rabies immunoglobulin by IM injection on D0 [93]. After 0, 14, 28, 42, and 6 months, their immunogenicity was evaluated and seroconversion rates were calculated as the rate of participants containing neutralizing antibody titers ≥0.5 IU/mL against rabies virus [93]. The geometric mean titers increased with antigen content at each time point. For the geometric mean titers of PVRV-NG2, the high-dose was the highest; however, the medium dose was similar to those with HDCV, and the dose was similar to PVRV-NG [93]. For the safety assay, PVRV-NG2 was similar to PVRV-NG, but PVRV-NG2 or PVRV-NG (36.7–47.5%) has fewer injection site reactions, compared to HDCV (61.5%) [93]. This study shows that the immunogenicity and safety profiles of the high-dose PVRV-NG2 can be a better vaccine for post-exposure prophylaxis than current HDCV vaccines.

In 2023, Huang et al. developed a Vero cell-based rabies vaccine (PVRV-WIBP) and evaluated the safety and immunogenicity for human use [94]. In the phase III clinical trial, a cohort of 40 participants at stage 1 and 1956 subjects at stage 2 aged from 10 to 50 years was recruited. For safety assessment, 20 participants at stage 1 received either four-dose or five-dose regimen of PVRV-WIBP [94]. However, at stage 2, 1956 subjects were randomly divided into three groups who received the five-dose PVRV-WIBP, five-dose PVRV-LNCD, and four-dose PVRV-WIBP, respectively [94]. They measured the serum neutralizing antibody titer against rabies on day 7 or 14 and day 35 or 42 post vaccination and recorded adverse events for more than 6 months. In the PVRV-WIBP (4 and 5 doses) and PVRV-LNCD groups, they found most adverse reactions were mild and even moderate reactions were resolved within 1 week after each injection [94]. The results revealed that all three groups had complete seroconversion 14 days after the initial dose [94]. Compared to those in the PVRV-LNCD group, the susceptible subjects in the PVRV-WIBP group (four-dose or five-dose) demonstrated higher neutralizing antibody titers against the rabies virus 14 days after completing the full vaccination schedule [94]. PVRV-WIBP induced almost the same immune responses as PVRV-LNCD and was well tolerated in healthy individuals aged 10–50 years [94]. This study shows that PVRV-WIBP (both four- and five-dose schedules) can be an alternative rabies vaccine for post-exposure prophylaxis.

#### 3.3.2. Post-Exposure Prophylaxis Using Rabies Vaccines Based on Vero Cells and Human Rabies Immunoglobulins

The WHO recommends extensive wound washing, immediate vaccination, and administration of human blood-derived rabies immunoglobulins (HRIG) in severe rabies category III exposures for post-exposure prophylaxis [95]. USA Center for Disease Prevention and Control (CDC) also suggests that rabies post-exposure prophylaxis includes wound washing, human rabies immune globulin (HRIG), and a four-dose series of vaccines [96]. HRIG is given only once at the beginning of post-exposure and only to previously unvaccinated persons. HRIG can provide immediate antibodies protection until the patients can actively producing antibodies of their own, but some studies have shown that HRIG may interfere with rabies vaccine immunogenicity to some extent [97,98]. Despite of this, the combination treatment of PVRV-NG and HRIG has demonstrated to be promising in some animal studies and clinical trials.

In 2022, Bernard et al. investigated the interference of HRIG on a next generation rabies vaccine candidate (PVRV-NG) based on Vero cells, compared to standard-of-care vaccines in a hamster model [99]. They evaluated the interference of either human or equine HRIG on the immune response induced by PVRV-NG, Verorab^®^ and Imovax^®^ Rabies (HDCV) by the four-dose post-exposure prophylaxis schedule [99]. For the vaccines administered with or without HRIG, they also determined the seroneutralizing antibody titers against rabies and specific serum IgM titers, respectively [99]. They transiently found HRIG interference with PVRV-NG, which was similar to that in PVRV and tended to be lower than HDCV on day 7 [99]. The results revealed that HRIG elicit similar or less interference on PVRV-NG than the standard-of-care vaccines in hamsters [99]. The study shows that the treatment, which combines PVRV-NG based on Vero cells and HRIG, is perspective for the post-exposure prophylaxis of rabies in humans.

In 2024, Pineda-Peña et al. developed a next-generation rabies vaccine (PVRV-NG2) using the same Pitman–Moore strain as in the licensed PVRV (Verorab@) and HDCV (Imovax Rabies^®^) [100]. They evaluated the immunogenicity and safety of PVRV-NG2 with and without HRIG IM injection to compare with PVRV+HRIG and HDCV+ HRIG in a similar post-exposure prophylaxis model [100]. In the dual-center, modified, and double-blind phase III study, healthy adults ≥18 years old (N = 640) were randomized to PVRV-NG2+HRIG, PVRV+HRIG, HDCV+HRIG, or PVRV-NG2 alone by the proportion 3:1:1:1 and received as single vaccine on days D 0, D3, D7, D14, and 28, with HRIG on D0 in applicable groups [100]. They determined RVNA titers on D0 and D14, D28, and D42 (post-vaccination). All participants achieved rabies virus neutralizing antibody (RVNA) titers ≥0.5 IU/mL (primary objective) to show non-inferiority of PVRV-NG2+HRIG compared with PVRV+HRIG and HDCV+HRIG [100]. They also assessed the adverse events up to 6 months after the last injection. The results revealed that almost all participants (99.6%, PVRV-NG2+HRIG; 100%, PVRV+HRIG; 98.7%, HDCV+HRIG; 100%, PVRV-NG2 alone) achieved RVNA titers ≥0.5 IU/mL at D28 and the geometric mean titers were similar between groups at all time points [100]. Moreover, the safety profile of the vaccine was similar between PVRV-NG2 and other groups [100]. The study demonstrates that the immunogenicity and safety of PVRV-NG2+HRIG are almost the same as current standard-of-care rabies vaccines for post-exposure prophylaxis, and thereby PVRV-NG2+HRIG is promising to be a combination treatment for human rabies.

#### 3.3.3. Post-Exposure Prophylaxis Using Rabies Vaccines Containing Adjuvants

In 2023, Yu et al. used PIKA, a chemically stabilized analog of double-stranded (ds) RNA that could interact with toll-like receptor 3 (TLR3), as an adjuvant in the inactivated purified rabies virus vaccine, to promote immune responses [70]. In the animal study, they tested seven rabies viral strains prevalent in China. The results demonstrated that the PIKA rabies vaccine had more than 80% protective efficacy without the use of immunoglobulins in mice [70]. The PIKA rabies vaccine showed that the neutralizing antibody levels were significantly enhanced within 5 days post-vaccination, indicating more immune responses were induced than with licensed rabies vaccines [70]. They found that the PIKA rabies vaccine resulted in a significant increment in T cells that produce IFN-γ responding to the antigen [70]. Furthermore, the levels of IL-1β, IL-6, CCL-2, and TNF-α were all elevated at the injection site following administration of the PIKA rabies vaccine [70]. The results also suggested that the levels of chemotactic proteins and pro-inflammatory molecules were increasing in the serum [70]. The mode of action of PIKA was further confirmed by the test in TLR3-knockout mice, verifying its function was dependent on the TLR3 pathway [70]. The study shows that the PIKA has the potential to be an adjuvant to result in a significantly high efficacy of rabies vaccines without HRIG administration.

#### 3.3.4. Post-Exposure Prophylaxis Using Monoclonal Antibody Combinations

The passive immunization using immunoglobins for human rabies post-exposure prophylaxis is either human monoclonal antibody (mAb) or HRIG. Currently, the replacement of traditional HRIG with emerging mAb or mAb combinations is recommended because of the limited supply and potential safety risks of HRIG.

In 2024, Long et al. developed a combination mAb CRM25 by mixing two human mAbs, RM02, and RM05 (mass ratio 1:1) which were non-competing and non-overlapping mAbs [101]. They found that CRM25 could cross-neutralizing rabies virus strains and additionally demonstrated an inhibitory effect on the infection of all tested common rabies viruses and non-rabies virus phylogroup I lyssaviruses [101]. The results revealed that CRM25 could protect Syrian golden hamsters from lethal rabies virus challenges, compared to HRIG [101]. The study shows that CRM25 may be a potential therapeutic candidate for rabies post-exposure prophylaxis in future clinical trials.

## 4. Discussion

Rabies is transmitted mainly by animal bites (especially dogs); thus, pre-exposure prophylaxis is crucial for animals. Because humans are not rabies vectors. Except in cases of organ transplants, pre-exposure prophylaxis is only required for professional risk and post-exposure prophylaxis is critical for preventing rabies in human cases. The development of animal vaccines should be focused on pre-exposure prophylaxis. However, unlike animal rabies vaccines, the development of human rabies vaccines is focused on the clinical application for post-exposure prophylaxis. For the goal of the WHO to eradicate rabies by 2030, the developments and advantages of the new rabies/lyssaviruses vaccines should be highlighted.

The challenges of traditional animal vaccines, including insufficient coverage, safety, questionable long-term effectiveness, and resistance, may be obstacles for animal rabies promotion. Fortunately, these limitations can be overcome by some novel vaccine candidates; for example, an inactivated vaccine that is relatively safe with an adjuvant is able to enhance effectiveness; an oral attenuated vaccine is able to enlarge vaccination coverage; an mRNA-based vaccine potentially offers a safe, effective, and less resistant vaccine. Therefore, orally administered mRNA-based vaccines can be considered a potential animal rabies vaccine in the future, though the cold chain requirement is still a critical issue, especially for wild animals.

Current human rabies vaccines are usually considered therapeutic vaccines for pathogens because they are usually used for post-exposure immunization against rabies viruses. Prompt treatment after exposure to rabid animals is critical in that the rabies vaccine is not effective against viral infection that has already developed clinical symptoms, especially after invading the CNS. Current human rabies vaccines (e.g., HDCV, HKCV, PCECV) have some disadvantages including incomplete immune responses, side effects, a multiple dose requirement, and expense. Therefore, PVRV has become a suitable object for vaccine development to enhance efficacy and safety, as well as reduce expense. The advantages of Vero cells include many cycle divisions without aging, having no animal or human components and low residual DNA content. Additionally, Vero cells are easy to be cultivated in medium at low cost and can be purified and freeze-dried for the production of rabies vaccines. To enhance the efficacy, the combination of PVRV and HRIG may be needed. However, HRIG is usually difficult to harvest and expensive, and may even interfere with rabies vaccine immunogenicity to some extent in some studies. In contrast to HRIG, an adjuvant (e.g., PIKA) and mAb are more easy to obtain, cheap, and secure. Therefore, the addition of an adjuvant or mAb combination may be an alternative choice for rabies vaccines without HRIG administration.

In addition to PVRV, the mRNA-based vaccine is another promising human rabies vaccine candidate, though it is required to have more evidence to prove both its safety and effectiveness in clinical trials [102]. The mRNA-based vaccine has several significant advantages as follows: (1) safe and less resistant, because only a small portion of the viral genetic material is used; (2) the dose rate is low; (3) the lowest possibility that the rabies virus will reverse itself; (4) low risk of insertional mutagenesis; (5) high potency; (6) accelerated development cycles; (7) potential for low-cost manufacture [103,104]. Moreover, the mRNA-based vaccine has been shown to be effective to induce both cellular immunity and humoral immunity against rabies in some animal studies [105,106,107].

Current mRNA vaccines for human uses are mainly given by intramuscular or subcutaneous injection. For the reduction of pain, stress, cost, and the professional requirement of vaccinations, non-invasive routes including oral and aerosol delivery (e.g., intranasal and oral inhalation) are becoming the preferred routes for animal vaccines. Therefore, new vectors should be designed to improve the efficacy and stability of mRNA vaccines for these non-invasive administration routes. Scientists have to evaluate the adverse events and allergic reactions resulting from mRNA vaccines in natural animals. Additionally, there is a need to develop a safe and effective delivery tool (e.g., nanoparticle, virus, virus-like particle) that protects mRNA vaccines from digestion and denaturation in animals. At present, the expense of mRNA vaccines is still high due to the requirement of expensive delivery tools and the cold chain. In case the mRNA vaccine can be developed at a low cost and high quality in the future, it is extremely important for the prevention of both human and animal rabies. 

## Figures and Tables

**Figure 1 microorganisms-13-00380-f001:**
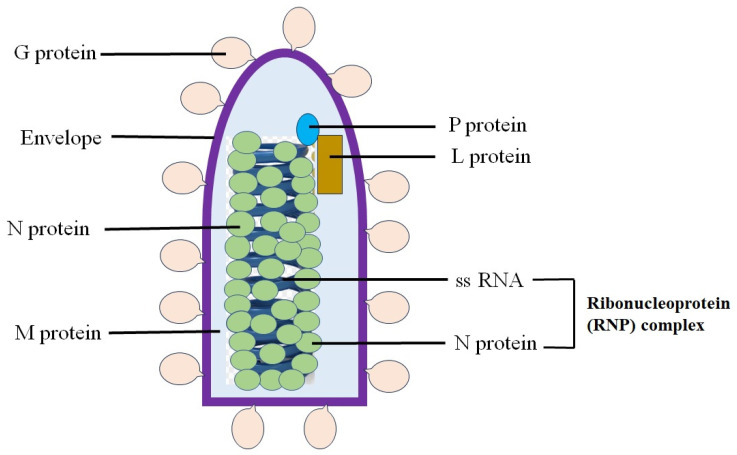
Structure of a rabies virus: The nucleoprotein (N) coats the linear single-stranded RNA (ssRNA) to form a nucleocapsid with helical symmetry. The phosphoprotein (P) and the large structure protein (L) are associated with the nucleocapsid. The L protein is multifunctional, and its gene takes up about half of the viral genome. The matrix (M) protein forms a layer between the nucleocapsid (nucleoprotein) and the envelope, which is the outermost layer of the virion. The glycoprotein (G) forms knob-like spikes that protrude from the envelope.

**Table 1 microorganisms-13-00380-t001:** Types and characteristics of traditional vaccines for animal rabies.

Type	Characteristics	References
Inactivated vaccine	The inactivated vaccine is produced using physical or chemical technology, such as chemicals: hydrogen peroxide, binary ethylenimine, ethylenimine, etc. The inactivated vaccine maintains the viral immunogenicity without causing pathogenicity. The inactivated vaccine can produce a specific immune response in the animal to prevent the infection of the rabies virus after vaccination.	[48,49,50]
Live attenuated vaccine	The live attenuated vaccine is a rabies virus that has virulence reduced to make it non-pathogenic but still immunogenic. The live attenuated vaccine produces a more effective and relatively long-lasting immune protection in animals. The live attenuated vaccine contains live viruses, so the vaccination needs to be strictly controlled when it is used to refrain the live viruses from causing reverse pathogenicity.	[51,52,53]
Recombinant vaccine	The recombinant vaccine uses genetic engineering technologies to insert some gene fragments of the rabies virus into harmless vectors, allowing these vectors to produce an immune response similar to the rabies virus in animals. The advantages of a recombinant vaccine include good immune effects, low production costs, and high safety which can avoid the risk of vaccine causing diseases.	[54,55,56,57]

**Table 2 microorganisms-13-00380-t002:** The limitations of traditional vaccines for animal rabies.

Limitations	Description	References
Insufficient coverage	The animal vaccination coverage is still low because of traffic, expense, and administration inconvenience, especially in some low-income and resource-limited countries. Economic conditions and public health infrastructure in some regions limit the popularization of animal vaccines, resulting in great challenges in rabies prevention and control in animals. For herd immunity, it is necessary to have an adequate approximation of the number of animals that should be vaccinated, but this is difficult to evaluate and not affordable in many countries.	[59,60,61]
Safety concern	The individual animals may experience hypersensitivity or adverse reactions. Especially, live attenuated vaccines may have reverse or spontaneous mutation to cause animal diseases, and even lead to environmental pollution.	[52,62]
Questionable long-term effectiveness	The long-term efficacy and potency of animal vaccines are a question, especially for inactivated vaccines.	[63,64]
Induced resistance against rabies viruses is variable	The antibody production is controlled by polygenes, and the rabies vaccine may induce different resistance to viruses in high and low antibody responder animals.	[65]
Oral resistance	The uptake efficiency of rabies vaccines is variable in different host species in the wild. The wild environment is hostile, and proteases are abundant in the gastrointestinal tract of animals.	[66,67]
Cold chain requirement	The maintenance of cold chain for the shipment and storage of animal rabies vaccines is difficult, especially in remote areas where there is no electricity.	[68,69]

**Table 3 microorganisms-13-00380-t003:** Types and characteristics of current vaccines or immunoglobulins for human rabies.

Type	Characteristics	References
Cell culture vaccine (CCV)	CCV is the most common rabies vaccine used in human and has been widely applied clinically. CCV is made by rabies virus produced in cell culture with high purity and low side effects. CCV is safe and stable, containing no toxic by-products and widely used in many countries for post-exposure prophylaxis and suitable for both adults and children.	[75,76,77,78,79,80,81,82]
Serum	The anti-rabies serum is obtained by purifying hyperimmune serum of healthy equines having specific activity of neutralizing the rabies virus. The anti-rabies serum is used in conjunction with vaccines after infection with rabies viruses, especially for severe exposures as deep bites patients who had not never received antiserum and vaccine.	[83]
Immunoglobulin	Rabies immunoglobulins are given to persons who have been exposed to a rabid animal as post-exposure prophylaxis. Rabies immunoglobulins are used only in persons who have never received the rabies vaccine. Rabies immunoglobulins provide direct antibody protection for post-exposure emergency use, promoting patients to be effectively protected before autoantibodies are produced in the body.	[84,85,86]

**Table 4 microorganisms-13-00380-t004:** The disadvantages of current vaccines for human rabies.

Disadvantage	Description
Incomplete immune response	Because of delayed antibody response and weak cellular immunity, some people’s immune systems may not produce enough antibodies and provide complete cellular immunity after vaccination, resulting in insufficient immune responses [87]. The correlation factors include individual difference, vaccine type, vaccination method, and physical condition at the vaccination time.
Side effect	Local reaction: Redness, swelling, pain, or a hard lump may appear at the vaccination site. Systemic reaction: It is usually mild and will disappear within a few days, such as fever, headache, fatigue, nausea, etc. Anaphylaxis: It is rare, but some people may have an allergic reaction to ingredients in the rabies vaccine which in the most severe cases may lead to anaphylactic shock [87].
Multiple dose vaccination requirement	It may be troublesome and inconvenient for some people that vaccination requires multiple doses for optimal effectiveness, especially if the vaccine is given after exposure. For those who have been exposed to rabies viruses, vaccination must be administered as early as possible. It may be required to use a combination of rabies vaccine and immunoglobulin, further increasing the complexity and cost of treatment [87].
Expense	Although rabies vaccines are free or inexpensive in most areas, the cost of the vaccine can be a significant burden in some low-income countries or regions [87]. Because rabies vaccination requires multiple doses, some people may not be able to complete the vaccination in time due to financial reasons.
Insufficient vaccine supply	The supply of rabies vaccine may be limited in some areas, especially if there are problems with vaccine production or distribution, which may affect the timely supply of the vaccine [87].
Cold chain requirement	Because rabies vaccines must be keep in refrigeration or freeze, cold chain maintenance can be challenging especially in resource-limited areas or some localities without electricity [88,89].

## Data Availability

Not applicable.

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
