# Peer review of "Infection and Prevention of Rabies Viruses"

_microorganisms, 2025, doi:10.3390/microorganisms13020380_

Round 1

Reviewer 1 Report

Comments and Suggestions for Authors

The authors reviewed rabies viruses and rabies vaccines. Rabies is an important zoonotic and fatal disease. Reviewing the current research about rabies is valuable for readers. However, the present manuscript has too many problems. Specific comments are as follows.

L43: Rabies virus has one layer of envelope. "outer envelope" should be "envelope". Delete "outer".

L43-44: Add an explanation that the rabies virus genome is a linear RNA.

L46: Add "(N)" after "nucleoprotein".

L55-56: Figure 1 is not cited in the text. Where is Figure 1 sited?

Figure 1 art is wrong and poor. Redraw the scheme of rabies virus particle which is described like a bullet. And the viral RNA is a single stranded linear RNA surrounded by N, P, and L proteins to form a helical nucleoprotein complex. M proteins are under envelope. Hopefully, an electron microscopic picture should be shown added to the scheme.

L64-65: The nicotinic acetylcholine receptor is in the skeletal neuromuscular junction, not host skeletal muscle cell membrane. Correct the description.

L75 The description "depending on different host individuals" is not right and inaccurate. The incubation periods vary depending on the distance from the bite lesion sites to the brain.

L76-78: The authors described "During this period, P protein .... invading viruses [14, 15]". This description is not correct. P protein acts as an interferon antagonist throughout a replication period of rabies virus in infected cells and does not relate to the incubation period. Delete this sentence or correct the description.

And, the reference 14 does not report P protein as an interferon antagonist. Delete this citation.

L85: The reference 85 reported enteric viruses like norovirus, rotavirus and astrovirus replication in salivary glands and not anything about rabies virus. Delete this citation and cite appropriate references.

L86-96: References 18 to 21 are review articles and not original reports. If this manuscript is a review, the authors should cite the original articles. 

L97: The authors describe "gasdermin D (Gsdmd) gene of rabies viruses". This is a wrong description. "gasdermin D (Gsdmd) is one of the host genes and not a virus gene. Correct the description.

L103-104: A sentence "In the CNS, it will cause... and brainstem" is hard to understand. Does it mean the CNS damage caused by rabies virus will cause paralysis and patients will appear depressed and so on? Correct the description much more clearer.

L105-105: References 10, 21 and 23-25 are review articles and a web site. The web site 25 does not exist and has changed its URL. Cite a correct URL with the date referenced. Other original articles should be cited.

L109-112: This description is wrong. The authors described that the incubation period is usually 1 t 3 months in L106. "Symptoms usually appear within a few days to a week after infection" is the wrong description. Correct and describe scientifically.

L119-122: The authors should describe how long to continue these symptoms until the patient death clearly.

L138-141: The reference 29 does not include data about illustration of the major vectors. Cite an appropriate reference.

L147-148: The reference 32 is not a report by WHO. Cite a correct reference.

L152-167: Meaning of this paragraph is unclear. First, it should be described that Rabies virus is genotype 1 out of 7 genotypes of Lyssaviruseso. It is better to show a table explaining the seven genotypes of Lyssaviruses. Furthermore, if the authors discussed the vaccination for bats and other animals for preventing rabies or lyssavirus infections, immunological relatedness among the seven genotypes of Lyssaviruses should be described in this paragraph. It should be divided into descriptions of the rabies virus and those of the other lyssaviruses.

L184-185: Description " The first rabies vaccine which is a live attenuated vaccine was developed by .." is wrong. The vaccine by Dr. Pasteur was dried homogenate of rabbit spinal cord and inactivated vaccine. After several inoculations with the inactivated vaccine, Dr Pasteur inoculated a homogenate of one day-dried spinal cord which might include live viruses. Correct the description and cite an appropriate reference.

L186-188: Add a description that all of these vaccines are inactivated vaccine.

L189-191: A live attenuated vaccine is used for wild animal vaccination and not used commonly for domestic and companion animals. Therefore, the live attenuated vaccine should not be described separately.

Historically, the inactivated vaccines have been widely used and then the live attenuated vaccine for wild animals is used for controlling rabies in wild animals. The order of the description should be changed and clarified.

L192-193: In Table 1, inactivated vaccine should be the first and then live attenuated vaccine and recombinant vaccine. Citations are wrong. Reference 38 is too old to be cited. References 39 to 41 are reports about recombinant vaccines. The characteristics of the inactivated vaccine are also wrong. The inactivated vaccine is produced using chemical treatment such as beta-propiolactone and not radiation. Correct descriptions.

L194-206: Description about traditional vaccine challenges lacks details about actual protection efficiency by the traditional vaccines. Citations are poor and wrong in this paragraph, too. Rewrite this paragraph completely.

L207-208: What does "challenges" mean in Table 2 and this paragraph? Clarify the meaning of "challenges".

The description about safety concerns is unclear and not scientific. Safety of vaccines is related to inactivated or live-attenuated and not the traditional or others. Rewrite the description.

The authors did not understand the meaning of "resistance" in references 51-53. "Drug resistance or oral resistance" is not related to data shown in these references. Correct the description of "Resistance".

L220-230: The authors described data in reference 54. However, there are many reports about adjuvant for rabies vaccine. This paragraph is not enough. Comparative description about adjuvant for rabies vaccine is required.

L244-246: The oral rabies vaccination for wild animals has been examined  for a long time in various areas in the world. What is new in this paragraph?

L288-290: The authors described that current rabies vaccines ... are mainly divided into three types (Table 3). However, two types of vaccine and two types of immunoglobulins are shown in Table 3. And, no explanation about the three types in the text. Add an explanation about three types and clarify the text and Table 3.

Furthermore, what are the differences between CCV and PCECV? Both are cell culture vaccines. If the difference is what cells are used, both are the same cell culture vaccine and there is no need to distinguish them.

L293-295: Add an explanation about disadvantages in the text. Showing the table is not enough.

L296-297: Add references based on the descriptions.

L435-497: "Conclusion" is redundant. The description has to be revised to be compact and clearer.

Comments on the Quality of English Language

No comments.

Author Response

The authors reviewed rabies viruses and rabies vaccines. Rabies is an important zoonotic and fatal disease. Reviewing the current research about rabies is valuable for readers. However, the present manuscript has too many problems. Specific comments are as follows.

 L43: Rabies virus has one layer of envelope. "outer envelope" should be "envelope". Delete "outer".

Ans: We have deleted “outer” to change “outer envelope” to “envelope”. (P.1, line 43)

L43-44: Add an explanation that the rabies virus genome is a linear RNA.

Ans: We have added an explanation that the rabies virus genome is a linear RNA. (P.1, line 44)

L46: Add "(N)" after "nucleoprotein".

Ans: We have added “(N)” after “nucleoprotein”. (P.2, line 46)

L55-56: Figure 1 is not cited in the text. Where is Figure 1 sited?

Ans: Figure 1 was originally cited in the text and we hightlighted it in yellow now. (P.2, line 49)

Figure 1 art is wrong and poor. Redraw the scheme of rabies virus particle which is described like a bullet. And the viral RNA is a single stranded linear RNA surrounded by N, P, and L proteins to form a helical nucleoprotein complex. M proteins are under envelope. Hopefully, an electron microscopic picture should be shown added to the scheme.

Ans: We have redraw the the scheme of rabies virus particle, but we cannot draw an electron microscopic picture because we have no rabies virus and electron microscope. (P.2, line 46)

L64-65: The nicotinic acetylcholine receptor is in the skeletal neuromuscular junction, not host skeletal muscle cell membrane. Correct the description.

Ans: We have corrected the description into “skeletal neuromuscular junction”. (P.3, line 67)

L75 The description "depending on different host individuals" is not right and inaccurate. The incubation periods vary depending on the distance from the bite lesion sites to the brain.

Ans: We have changed the description to “depending on the distance from the bite lesion sites to the brain”. (P.3, line 81-82)

L76-78: The authors described "During this period, P protein .... invading viruses [14, 15]". This description is not correct. P protein acts as an interferon antagonist throughout a replication period of rabies virus in infected cells and does not relate to the incubation period. Delete this sentence or correct the description.

Ans:We have changed the statement to “P protein can acts as an interferon antagonist throughout a replication period of rabies virus in infected cells to reduce the host innate immunity to against invading viruses.” (P.3, line 82-85)

And, the reference 14 does not report P protein as an interferon antagonist. Delete this citation.

Ans: We have cited a new reference [15, 16] to report P protein as an interferon antagonist. (P.3, line 85)

L85: The reference 15 reported enteric viruses like norovirus, rotavirus and astrovirus replication in salivary glands and not anything about rabies virus. Delete this citation and cite appropriate references.

Ans: We have deleted the reference and cited appropriate references [18, 19].(P.3, line 92)

L86-96: References 18 to 21 are review articles and not original reports. If this manuscript is a review, the authors should cite the original articles. 

Ans: We have replaced the review references with original article references [20-22].(P.3, line 96)

L97: The authors describe "gasdermin D (Gsdmd) gene of rabies viruses". This is a wrong description. "gasdermin D (Gsdmd) is one of the host genes and not a virus gene. Correct the description.

Ans: We have changed the the description to “The recent study also suggested that host gasdermin D (Gsdmd) genes related to the pyroptosis (a highly inflammatory form of lytic programmed cell death that occurs often upon an intracellular rabies virus infection) pathway were significantly upregulated to cause the neuron damage.” (P.3, line 104-107)

L103-104: A sentence "In the CNS, it will cause... and brainstem" is hard to understand. Does it mean the CNS damage caused by rabies virus will cause paralysis and patients will appear depressed and so on? Correct the description much more clearer.

Ans: We have corrected the sentence into “The CNS damage caused by rabies virus will cause paralysis and patients will appear depressed and so on.” (P.3, line 111-113)

L105-105: References 10, 21 and 23-25 are review articles and a web site. The web site 25 does not exist and has changed its URL. Cite a correct URL with the date referenced. Other original articles should be cited.

Ans: We have cited a new web site 28 to instead of 25 and replaced the review articles with original articles as references [28-31]. (P.3, line 114)

L109-112: This description is wrong. The authors described that the incubation period is usually 1 t 3 months in L106. "Symptoms usually appear within a few days to a week after infection" is the wrong description. Correct and describe scientifically.

Ans: We have changed the sentence to “Symptoms start to appear after incubation period, including fever, tiredness, headache, tingling, numbness, burning at the injured site, malaise, loss of appetite, and sometimes flu-like symptoms. The duration of this period is 2-10 days.”(P.4, line 118-122)

L119-122: The authors should describe how long to continue these symptoms until the patient death clearly.

Ans: We have revised this paragraph and added 3 references [32-34]. (P.4, line 129-130)

L138-141: The reference 29 does not include data about illustration of the major vectors. Cite an appropriate reference.

Ans: We have deleted reference 29 and used a new reference [39]. (P.4, line 153)

 L147-148: The reference 32 is not a report by WHO. Cite a correct reference.

Ans: We have cited a correct reference [33]. (P.4, line 164)

L152-167: Meaning of this paragraph is unclear. First, it should be described that Rabies virus is genotype 1 out of 7 genotypes of Lyssaviruseso. It is better to show a table explaining the seven genotypes of Lyssaviruses. Furthermore, if the authors discussed the vaccination for bats and other animals for preventing rabies or lyssavirus infections, immunological relatedness among the seven genotypes of Lyssaviruses should be described in this paragraph. It should be divided into descriptions of the rabies virus and those of the other lyssaviruses.

Ans: We have replaced genotypes with species, describe the 18 species of Lyssaviruses and added a reference https://ictv.global/report/chapter/rhabdoviridae/rhabdoviridae/lyssavirus. Additionally, the immunological relatedness among the Lyssaviruses species has been described in this paragraph include more adequate and recent references [41-44]. (P.5, line 168-205)

 L184-185: Description " The first rabies vaccine which is a live attenuated vaccine was developed by .." is wrong. The vaccine by Dr. Pasteur was dried homogenate of rabbit spinal cord and inactivated vaccine. After several inoculations with the inactivated vaccine, Dr Pasteur inoculated a homogenate of one day-dried spinal cord which might include live viruses. Correct the description and cite an appropriate reference.

Ans: We have revised the live attenuated vaccine to dried homogenate of rabbit spinal cord and an inactivated vaccine. (P.6, line 221-222)

L186-188: Add a description that all of these vaccines are inactivated vaccine.

Ans: We have added a description that all of these vaccines are inactivated. (P.6, line 224)

L189-191: A live attenuated vaccine is used for wild animal vaccination and not used commonly for domestic and companion animals. Therefore, the live attenuated vaccine should not be described separately.

Historically, the inactivated vaccines have been widely used and then the live attenuated vaccine for wild animals is used for controlling rabies in wild animals. The order of the description should be changed and clarified.

Ans: We have changed the order and clarified the description to “Historically, the inactivated vaccines have been wildly used for domestic and companion animals and the live attenuated vaccines are usually used for controlling rabies in wild animals.” (P.6, line 229-232)

L192-193: In Table 1, inactivated vaccine should be the first and then live attenuated vaccine and recombinant vaccine. Citations are wrong. Reference 38 is too old to be cited. References 39 to 41 are reports about recombinant vaccines. The characteristics of the inactivated vaccine are also wrong. The inactivated vaccine is produced using chemical treatment such as beta-propiolactone and not radiation. Correct descriptions.

Ans: We have changed the order of Table 1, deleted reference 38 and update references [46-55]. Moreover, we have removed radiation for the production of inactivated vaccines in Table 1. (P.6, line 230)

L194-206: Description about traditional vaccine challenges lacks details about actual protection efficiency by the traditional vaccines. Citations are poor and wrong in this paragraph, too. Rewrite this paragraph completely.

Ans: We have rewrited this paragraph and Table 2 and included more new reference [57-67]. (P.6, line 234-249)

L207-208: What does "challenges" mean in Table 2 and this paragraph? Clarify the meaning of "challenges".

The description about safety concerns is unclear and not scientific. Safety of vaccines is related to inactivated or live-attenuated and not the traditional or others. Rewrite the description.

Ans: We have changed the word “challenges” to “limitations” and revised the description of safety concerns.(P.7, (P.6, line 249)

The authors did not understand the meaning of "resistance" in references 51-53. "Drug resistance or oral resistance" is not related to data shown in these references. Correct the description of "Resistance".

Ans: we have separated resistance to “Induced resistance to rabies viruses is variable” and “oral resistance” and provide different description in Table 2. (P.7, line 249)

L220-230: The authors described data in reference 54. However, there are many reports about adjuvant for rabies vaccine. This paragraph is not enough. Comparative description about adjuvant for rabies vaccine is required.

Ans: We have illustrated two more reports including references [69, 70] about adjuvants for rabies vaccines to compare description about adjuvant for rabies vaccine. (P.8, line 274-297)

L244-246: The oral rabies vaccination for wild animals has been examined  for a long time in various areas in the world. What is new in this paragraph?

Ans: We have revised the final description to “This study demonstrates that the oral rabies vaccine has the potential to be an important tool to target hard-to-reach free-roaming dogs in mass dog vaccination program.” (P.8, line 312-315)

 L288-290: The authors described that current rabies vaccines ... are mainly divided into three types (Table 3). However, two types of vaccine and two types of immunoglobulins are shown in Table 3. And, no explanation about the three types in the text. Add an explanation about three types and clarify the text and Table 3.

Furthermore, what are the differences between CCV and PCECV? Both are cell culture vaccines. If the difference is what cells are used, both are the same cell culture vaccine and there is no need to distinguish them.

Ans: We have combined CCV and PECEV to CCV because they are both cell culture vaccines and current rabies vaccines are mainly divided into three types. Additionally, we have added explanation about three types and clarify the text and Table 3.(P.10, line 360-368)

 L293-295: Add an explanation about disadvantages in the text. Showing the table is not enough.

Ans: We have added some explanation about disadvantages in the text. (P.10, line 370-P.11, line 379)

L296-297: Add references based on the descriptions.

Ans: We have added some reference [86-88] in Table 4. (P.11, line 379)

L435-497: "Conclusion" is redundant. The description has to be revised to be compact and clearer.

Ans: We have changed “Conclusion” to “Discussion”. (P14, line 521-P.15, line 87)

Reviewer 2 Report

Comments and Suggestions for Authors

The present review proposes an update of knowledge about the infection mechanisms and prevention methods of rabies. The bibliographic review includes the most recent articles. The manuscript is well written. The most important problem is that the two topics it addresses are treated in a very asymmetric way. The second part, which corresponds to prevention, is much more in-depth with an extensive discussion of the different vaccines and I consider that it can be a very important contribution for those interested in the area. On the other hand, the first part corresponding to infection presents many aspects that must be improved, especially many epidemiological aspects. The authors include percentages of signs and lesions without mentioning which species they are referring to. They do not describe in the necessary depth the importance of bats in the transmission of the disease. The pathogenesis is also described in a very superficial way, although the same authors point out that there are data that are not known, they would have to do a more in-depth analysis of the results of the works presented in the bibliography.

Author Response

The present review proposes an update of knowledge about the infection mechanisms and prevention methods of rabies. The bibliographic review includes the most recent articles. The manuscript is well written. The most important problem is that the two topics it addresses are treated in a very asymmetric way. The second part, which corresponds to prevention, is much more in-depth with an extensive discussion of the different vaccines and I consider that it can be a very important contribution for those interested in the area.

Ans: We thanks for the reviewr’s comment.

On the other hand, the first part corresponding to infection presents many aspects that must be improved, especially many epidemiological aspects.

Ans: We have improved the description about infection and epidemiological aspects. (P1, line 33-P.5, line 205)

The authors include percentages of signs and lesions without mentioning which species they are referring to.

Ans: We have revise the description of signs and lesions. (P3, line 109-P.4, line 134)

They do not describe in the necessary depth the importance of bats in the transmission of the disease.

Ans: We have revised the paragraph to describe the importance of bats in the transmission of the rabies. (P5, line 168-205)

The pathogenesis is also described in a very superficial way, although the same authors point out that there are data that are not known, they would have to do a more in-depth analysis of the results of the works presented in the bibliography.

Ans: We have revised the paragraph to describe the pathogenesis of the rabies virus. (P3, line 65-92)

Reviewer 3 Report

Comments and Suggestions for Authors

It is a very well done literature review article, exposing all types of vaccines for both animals and humans.  By comparing the different vaccines, it clearly concludes which is the best vaccine that could be used in humans, and includes the reasons why it arrives at this conclusion.

Can a general discussion be made, what is read inside the conclusion? 

The objection I have to this article is that the conclusion seems to be a discussion and not really the conclusion they reach. I consider that from lines 472 to 470 would be the conclusion.

This review article has a different structure than the articles in which they do experimentation. In each of the items listed by the authors, they discuss why it is better or why it is not useful. Therefore, the conclusion is something belonging to the authors of the article and not again a review.

Author Response

It is a very well done literature review article, exposing all types of vaccines for both animals and humans.  By comparing the different vaccines, it clearly concludes which is the best vaccine that could be used in humans, and includes the reasons why it arrives at this conclusion.

Can a general discussion be made, what is read inside the conclusion? 

Ans: We have the section of “Discussion” instead of “Conclusion.” (P14, line 521-P.15, line 87)

The objection I have to this article is that the conclusion seems to be a discussion and not really the conclusion they reach. I consider that from lines 472 to 470 would be the conclusion.This review article has a different structure than the articles in which they do experimentation. In each of the items listed by the authors, they discuss why it is better or why it is not useful. Therefore, the conclusion is something belonging to the authors of the article and not again a review.

Ans: We have replaced the section of “Discussion” with “Conclusion.” (P14, line 521-P.15, line 87)

Round 2

Reviewer 2 Report

Comments and Suggestions for Authors

In the present form the  manuscript can be accepted

Author Response

We greatly appreciate the comments and suggestions of academic editor and reviewers. We have answered their question and revised the manuscript accordingly in tracked forms.

Reviewer 3 Report

Comments and Suggestions for Authors      

The changes made, improved the article

Author Response

(The authors gave the same response as above.)
